# Myocardial Glutathione Synthase and TRXIP Expression Are Significantly Elevated in Hypertension and Diabetes: Influence of Stress on Antioxidant Pathways

**Anastasia Sklifasovskaya \*, Mikhail Blagonravov \*, Madina Azova and Vyacheslav Goryachev**

Institute of Medicine, RUDN University, 6 Miklukho-Maklaya St, 117198 Moscow, Russia;
azova-mm@rudn.ru (M.A.); goryachev-va@rudn.ru (V.G.)

\* Correspondence: sklifasovskaya-ap@rudn.ru (A.S.); blagonravov-ml@rudn.ru (M.B.);
Tel.: +7-(495)-433-12-11 (A.S.); +7-(495)-434-95-24 (M.B.)

**Abstract:** Antioxidant protection is one of the key reactions of cardiomyocytes (CMCs) in response to myocardial damage of various origins. The thioredoxin interacting protein (TXNIP) is an inhibitor of thioredoxin (TXN). Over the recent few years, TXNIP has received significant attention due to its wide range of functions in energy metabolism. In the present work, we studied the features of the redox-thiol systems, in particular, the amount of TXNIP and glutathione synthetase (GS) as markers of oxidative damage to CMCs and antioxidant protection, respectively. This study was carried out on 38-week-old Wistar-Kyoto rats with insulin-dependent diabetes mellitus (DM) induced by streptozotocin, on 38- and 57-week-old hypertensive SHR rats and on a model of combined hypertension and DM (38-week-old SHR rats with DM). It was found that the amount of TXNIP increased in 57-week-old SHR rats, in diabetic rats and in SHR rats with DM. In 38-week-old SHR rats, the expression of TXNIP significantly decreased. The expression of GS was significantly higher compared with the controls in 57-week-old SHR rats, in DM rats and in the case of the combination of hypertension and DM. The obtained data show that myocardial damage caused by DM and hypertension are accompanied by the activation of oxidative stress and antioxidant protection.

**Keywords:** myocardial damage; thioredoxin interacting protein; glutathione synthetase; insulin-dependent diabetes mellitus; hypertension

## 1. Introduction

Antioxidant protection is one of the key reactions of cardiomyocytes (CMCs) observed in response to myocardial damage of various origins. The thioredoxin interacting protein (TXNIP) operates as an important regulator of carbohydrate and lipid metabolism through pleiotropic actions including the control of β-cell functions [1], gluconeogenesis, glucose uptake by peripheral tissues, adipogenesis and substrate utilization [2]. TXNIP binds and inhibits the activity of thioredoxin (TXN) affecting the redox balance of the cells, including CMCs [3,4]. The role of TXNIP in the pathogenesis of neurodegenerative diseases, such as Alzheimer's disease, has also been shown. In particular, TXNIP can be expressed by neurons, microglia, astrocytes and endothelial cells. It is also involved in neuronal death due to increased oxidative stress, inflammatory degeneration and vascular dysfunction [5]. Enhanced oxidative stress and production of reactive oxygen species (ROSs) suppress the endogenous tissue production of antioxidant substrates and impair cellular functions. Hyperglycemia stimulates ROS production [6]. At lower concentrations, ROSs also serve as secondary messengers regulating cell functions in the myocardium [7]. The redox states of the thiol systems are controlled by TXN, glutathione (GSH) and cysteine (Cys) [8,9]. An imbalance between ROS and antioxidant protection manifests

as aberrant cellular signaling and dysfunctional redox control in cells, which is particularly important for CMCs [10].

In most cells, GSH is synthesized de novo in a two-step reaction. First, γ-glytamyl-cysteine is formed from glutamate and cysteine catalyzed by glutamate cysteine ligase (GCL). Then glycine is added by glutathione synthetase (GS) to form active GSH [11]. At the same time, overexpression of glutathione synthetase increases the intracellular level of GSH [11,12].

Recent studies have shown that high glucose levels are accompanied by a significant increase in the TXNIP expression [13–15]. Overexpression of TXNIP in β-cells induced apoptosis and reduced insulin production. Meanwhile, TXNIP deficiency protected β-cells from apoptotic death, leading to an increase in insulin secretion [16,17]. It was also shown that deletion of the *TXNIP* gene in a streptozotocin-induced model of DM in mice resulted in improved glucose tolerance and was accompanied by the remission of hyperglycemia [14]. Moreover, H. Parikh et al. (2007) found that an inverse correlation between TXNIP expression and the rate of insulin-dependent glucose uptake in skeletal muscles is associated with a higher risk of type 2 DM in humans [15]. These results suggest that the downregulation of TXNIP in pre-diabetic and diabetic conditions may be beneficial for the treatment of human DM. So, taking into account the important role of TXNIP in glucose metabolism, this factor has been recognized as an attractive target for the treatment of type 1 and type 2 DM, as well as complications associated with cardiovascular diseases [18] and renal pathology [19,20].

Gao et al., (2020) determined the role of TXNIP as a regulator of metabolic switching during an ischemia-reperfusion (I/R) injury of the myocardium. It was also shown that TXNIP removal prevented cardiac dysfunction in response to pressure overload and I/R damage [21]. Another study demonstrated that the administration of exogenous TXN protected the heart from I/R injury, while TXNIP caused its exacerbation [22]. J. Chen et al., (2016), using a new TXNIP knockout mouse model, found that TXNIP deficiency contributed to the β-oxidation of fatty acids in CMCs via signal transduction through a specific miRNA—miR-33a [23].

TXNIP has a variety of functions in a number of processes, such as apoptosis, regulation of ROS and inflammation, by acting as a scaffold for several proteins [24]. TXNIP is also involved in cell signaling, and its stability is regulated by multiple mechanisms. In this regard, it is considered as a potential target for the pharmacotherapy of cardiovascular diseases [25]. According to the data obtained by G. Xiang et al., (2005), in mice with myocardial infarction, the expression of TXNIP increased. By using a DNA sequence-specific enzyme to downregulate the TXNIP mRNA, it was shown that CMC apoptosis was inhibited and cardiac function was improved [26]. Meanwhile, after myocardial infarction, TXNIP reduced HIF-1α and VEGF levels suppressed angiogenesis, enhanced apoptosis and ultimately determined a poor prognosis for the myocardium injury due to ischemia [4].

Despite the presence of a large amount of data describing the involvement of TXNIP and GS in cell responses to pathological factors of various diseases, there is no unambiguous idea of their role in the pathogenesis of myocardial damage caused by a combination of arterial hypertension (AH) and DM.

The objectives of our study were to assess the features of TXNIP and GS expression in CMCs from the left ventricular (LV) myocardium in AH, insulin-dependent DM and a combination of these two types of pathology.

## 2. Materials and Methods

### 2.1. Animals and Housing

The experiment was performed on 25 male rats, including 15 spontaneously hypertensive rats (SHR) and 10 Wistar-Kyoto (WKY) rats. The animals were obtained from the

Nursery for Laboratory Animals branch of the Shemyakin-Ovchinnikov Institute of Bioorganic Chemistry (Pushchino). Before the start of the research, the animals were maintained for 2 weeks in the laboratory in separate cages for acclimatization. After 2 weeks of adaptation, the experiment was carried out. During the experiment, each animal had free access to food (feeding time was at 7:00 p.m. daily) and water and was in an individual cage with a free-motion regime under the conditions of 12 h light-dark cycle with the room temperature maintained at +22–23 °C and relative air humidity of 40–50%. The rats with DM received a diet adapted for diabetic animals (dry complete diet food PRO PLAN ® Veterinary Diets DM St/Ox Diabetes Management for adult cats to regulate glucose intake (in diabetes mellitus) with low levels of sugars (mono- and disaccharides)). Other animals received the regular adapted rat food: laboratory feed for rodents (full-fledged extruded feed for keeping laboratory animals). The experiment was performed in compliance with the requirements of the European Convention for the Protection of Vertebrate Animals used for Experimental and Other Scientific Purposes (Strasbourg, 18.III.1986). In addition, the experiment was approved by the Ethical Committee of the RUDN Institute of Medicine (An ethical code number 26, date: 18 February 2021).

### 2.2. Experimental Design

The animals of the WKY and SHR genetic strains were divided into 5 groups: group 1 (control)—intact normotensive WKY rats aged 38 weeks; group 2—hypertensive SHR rats aged 38 weeks; group 3—hypertensive SHR rats aged 57 weeks; group 4—normotensive WKY rats aged 38 weeks with insulin-dependent DM duration 30 days; and group 5—hypertensive SHR rats aged 38 weeks with insulin-dependent DM duration 30 days. Each group included 5 animals.

### 2.3. Modeling of Insulin-Dependent DM

Modeling in the animals of groups 4 and 5 insulin-dependent DM was performed by a single dose of Streptozotocin (Alfa Aesar, Ward Hill, MA, USA) at 65 mg/kg animal body weight through intraperitoneal injection [27]. The Streptozotocin solution for injections was prepared using a sodium citrate buffer of pH 4.5 at + 4 °C just before the use of Streptozotocin [28]. The glucose level was measured in the blood from the tail vein by the glucometer "AccuChek Active" ("Roche Diabetes Care GmbH", Mannheim, Germany) after 3 days of Streptozotocin injection. For further experiments, only animals with a glucose blood level above 16 mmol/L were selected. The duration of DM in the animals of groups 4 and 5 was 30 days from the moment of glycemia verification [29].

### 2.4. Morphological and Immunohistochemical Study

Under general anesthesia, the animals of all the experimental groups were sacrificed, and a thoracotomy and extirpation of the heart were carried out. The obtained samples of LV myocardium were fixed for 72 h in 4% neutral paraformaldehyde. The material was processed and integrated into paraffin according to the standard method. Histological sections (5 μ) were sliced using Slidt 2003 microtome and attached to ordinary glasses for the morphological study and on poly-L-lysine-coated glasses for the immunohistochemical analysis.

Histological sections were examined using the Nikon Eclipse E400 light microscope at a 400× magnification and the TauVideo video system with the Tau Morphology program based on the Watec 221s camera. Morphometric analysis was performed in each field of vision (30 visual fields were analyzed in each myocardial preparation): the relative content of myofibrils, nuclei, vessels, destruction sites and intercellular spaces in the myocardium was determined in volume percentages (vol. %) using the Avtandilov grid: it was counted using the ratio of equally distant points occupied by the positively colored

cytoplasm of CMC to the total number of points occupied by the cytoplasm [30]. The "nuclear-cytoplasmic ratio" was also calculated: the percentage ratio of the area of the CMC nuclei to the area of muscle fibers.

According to the morphological analysis, in the group of SHR rats aged 38 weeks, compared with the control group, a significant increase in the volume area of myofibrils and a decrease in the number of nuclei were revealed. In addition, hypertrophy of muscle fibers was indicated by a decrease in NCR compared with the control. In the group of SHR rats aged 57 weeks, NCR was reduced relative to the control, which indicates hypertrophy of muscle fibers, however, it increased slightly relative to the group at 38 weeks, which is consistent with the results of previous studies: with long-term hypertension (more than a year), NCR does not differ significantly from the control group [31]. In the DM group, the volume fraction of CMCs decreased due to their hypotrophy, while the NCR (nuclear-cytoplasmic ratio) increased. In the AG group at 38 weeks, CMC death occurs in combination with DM, as evidenced by a sharp decrease in the number of nuclei, a return of the percentage of CMC to the control level and a decrease in NCR by 2 times relative to the control.

Slices from the immunohistochemical study were deparaffinized with xylene and processed in ethanol with descending concentrations. To assess the expression of TXNIP and GS in CMCs of the LV myocardium, an immunohistochemical reaction was performed with primary rabbit polyclonal antibodies—anti-TXNIP antibody produced in rabbit and anti-GS antibody produced in rabbit ("SigmaAldrich", St. Louis, MO, USA). The immunohistochemical reaction was visualized using the Rabbit-specific HRP/DAB (ABC) Detection IHC Kit (Abcam, Cambridge, UK). The slides were colored with Mayer's hematoxylin. The reaction was considered positive if a brown staining of the CMC cytoplasm was seen. Light microscopy was performed in each section of the LV myocardium in 30 fields of view at ×400 using a Nikon Eclipse E-400 microscope with the Watec 221S camera (Japan). Quantitative analysis of positively stained CMCs was carried out using an Avtandilov grid.

### 2.5. Statistics

The obtained data were processed using the Statistica 6.0 software (StatSoftInc., Tulsa, OK, USA). The average and standard error of the average were counted for each parameter. To determine the significance of the differences, the Mann–Whitney *U*-test was used (the difference between the average values was regarded as significant at $p \leq 0.05$).

## 3. Results

### 3.1. Immunohistochemical Study of TXNIP Expression in the LV Myocardium

According to the results of the quantitative analysis, in the experimental groups of isolated DM, AH in 57-week-old SHR rats and in the group of combined hypertension and DM, the content of TXNIP in the LV CMC cytoplasm was significantly increased compared with the controls (Figure 1). It is important to note that in the group of isolated DM, TXNIP expression increased by more than two times. On the contrary, in the group of 38-week-old hypertensive SHR rats, the protein expression decreased in comparison with the control group.

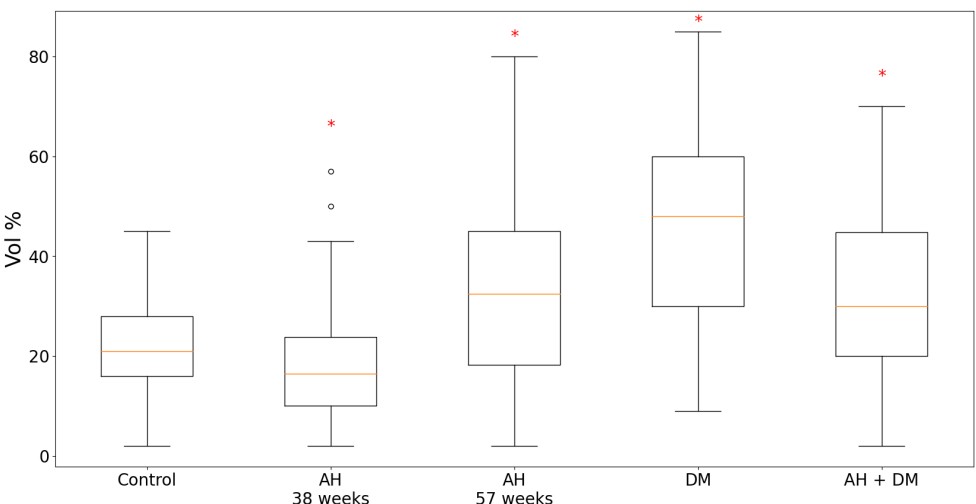

**Figure 1.** Content of TXNIP in LV CMCs in AH (38- and 57-week-old SHR rats), DM (38-week-old WKY rats with insulin-dependent DM) and the combination of AH and DM (38-week-old SHR rats with insulin-dependent DM). * — $p \leq 0.05$ in comparison with the control.

Qualitative analysis of the LV myocardial sections after the immunohistochemical reaction on TXNIP revealed the following picture. In the control group, continuous staining of high intensity was observed (Figure 2A). Both in the middle layer of the myocardium and towards the endocardium and epicardium, the density and intensity of positive staining remained. In the middle layer of the LV myocardium, there were some areas of local staining with medium intensity.

In the group of the 38-week-old hypertensive rats, the density of positively stained CMCs was significantly decreased compared with the controls. At the same time, the staining was less intense in comparison with the control group. Staining of medium intensity was predominantly seen towards the endocardium (Figure 2B). In the inner layer of the LV myocardium, local staining of CMCs of medium intensity was mainly visualized. Towards the epicardium, positive staining became local, but it had a higher intensity.

In the group of hypertensive animals aged 57 weeks, the density of CMCs with a positive reaction to TXNIP increased compared with the control group. Meanwhile, in the inner layer of the myocardium, the positive staining was continuous with medium intensity, and there were also separated intensively stained CMCs (Figure 2C). In the direction of the epicardium, the staining became local and had a higher intensity, which was typical for both individual CMCs and their groups. Towards the endocardium, positive staining became local too, but it had a high intensity against the background of weak continuous staining of most other CMCs.

In the case of isolated DM, the number of positively stained CMCs increased by two times compared with the group of intact animals. Moreover, in all the myocardial levels, continuous staining of high intensity was observed (Figure 2D). The CMC cytoplasm was intensively stained and some areas, with even more intense staining compared to the high degree of staining of the general background, were also seen.

For the group of hypertensive animals aged 38 weeks in combination with DM, an increase in the density of CMCs with a positive reaction to TXNIP was typical. At the same time, in the medium layer of the myocardium, against the background of general low intensity of staining, some individual intensively stained CMCs were found (Figure 2E). In the direction of the epicardium, there were some very intensively stained CMCs. Towards the endocardium, the intensity of the general background of the myocardium increased and some individual slightly stained CMCs were also seen.

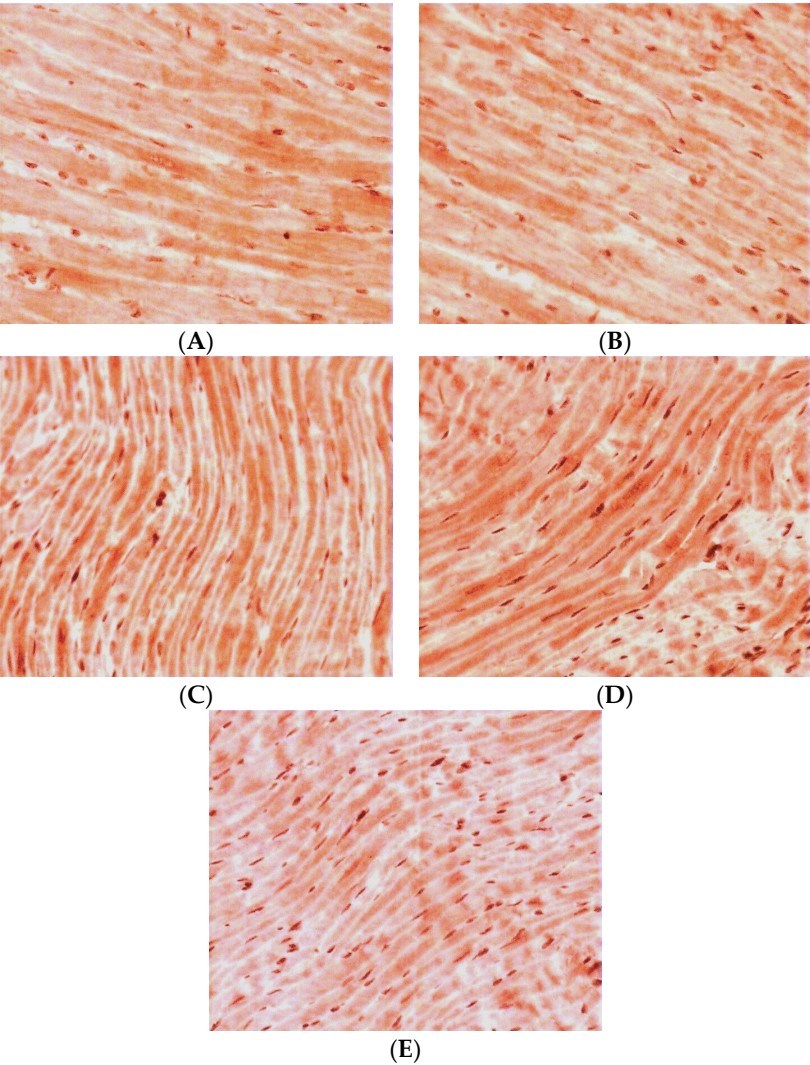

**Figure 2.** Expression of TXNIP in LV CMCs (the middle layer of the myocardium). Immunohisto-chemical staining, ×400. (**A**) control; (**B**) AH (38-week-old SHR rats); (**C**) AH (57-week-old SHR rats); (**D**) DM (38-week-old WKY rats with insulin-dependent DM); (**E**) AH+DM (38-week-old SHR rats with insulin-dependent DM).

*3.2. Immunohistochemical Study of GS Expression in the LV Myocardium*

According to the results of the quantitative analysis, in the animal groups of isolated DM and the combination of hypertension with DM, the expression of GS was significantly higher compared with the controls. Moreover, in the case of the isolated DM, the content of GS was twice as high (Figure 3). In the group of 38-week-old hypertensive rats, the expression of GS remained at the level of the control group, but in the 57-week-old SHR rats, it was slightly increased.

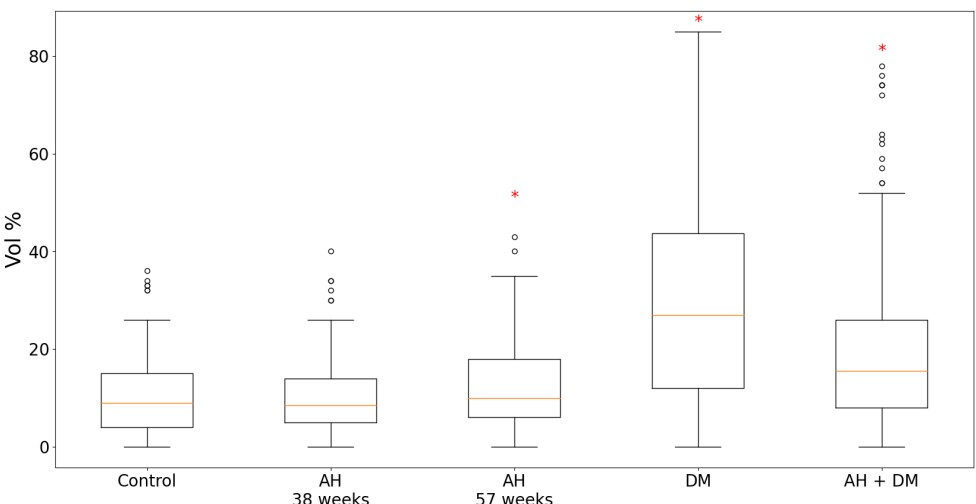

**Figure 3.** Content of GS in LV CMCs in AH (38- and 57-week-old SHR rats), DM (38-week-old WKY rats with insulin-dependent DM) and a combination of AH and DM (38-week-old SHR rats with insulin-dependent DM). * — $p \leq 0.05$ in comparison with the control.

Under qualitative evaluation of the LV myocardial sections after an immunohisto-chemical reaction on GS, the following features were observed. In the group of intact WKY rats, in the inner layer of the LV myocardium, continuous positive staining of low intensity was typical (Figure 4A). In the direction of the epicardium, CMC staining decreased and then disappeared completely. Towards the endocardium, a continuous staining with low intensity remained, and it was accompanied by some single CMCs with a medium intensity of staining.

In the group of SHR rats aged 38 weeks, the density of positive staining for GS remained at the level of the control group. In the inner layer of the myocardium, middle-intensity staining of only some groups of CMCs was observed (Figure 4B). The intensity of staining decreased and gradually disappeared completely towards the endocardium. In the direction of the epicardium, middle-intensity staining of some single groups of CMCs remained.

In the group of 57-week-old hypertensive animals, the number of CMCs with a positive reaction to GS was increased compared with the controls. In the inner layer of the myocardium, there was continuous staining with low intensity. Towards the epicardium and endocardium, some single groups of positively stained CMCs were seen (Figure 4C).

In the animals with isolated DM, in the middle layer of the myocardium, the intensity of positive staining and the number of CMCs with a positive reaction to GS markedly increased. In the inner layer of the myocardium, some groups of positively stained CMCs were found (Figure 4D). In the direction of the epicardium, groups of CMCs with a positive reaction to GS were seen more seldom. Towards the endocardium, the intensity of staining decreased. At first, low-intensity CMC staining was observed and then it completely disappeared.

For the group of 38-week-old animals with a combination of AH and DM, an increase of cells with a positive reaction to GS was characteristic. However, the density of positively stained CMCs was less compared with the group of isolated DM. Meanwhile, in the inner layer of the myocardium, continuous low-density staining was visualized. Towards the epicardium, some groups of positively stained CMCs were found (Figure 4E). In the direction of the endocardium, some single CMCs with positive middle-intensity staining were observed.

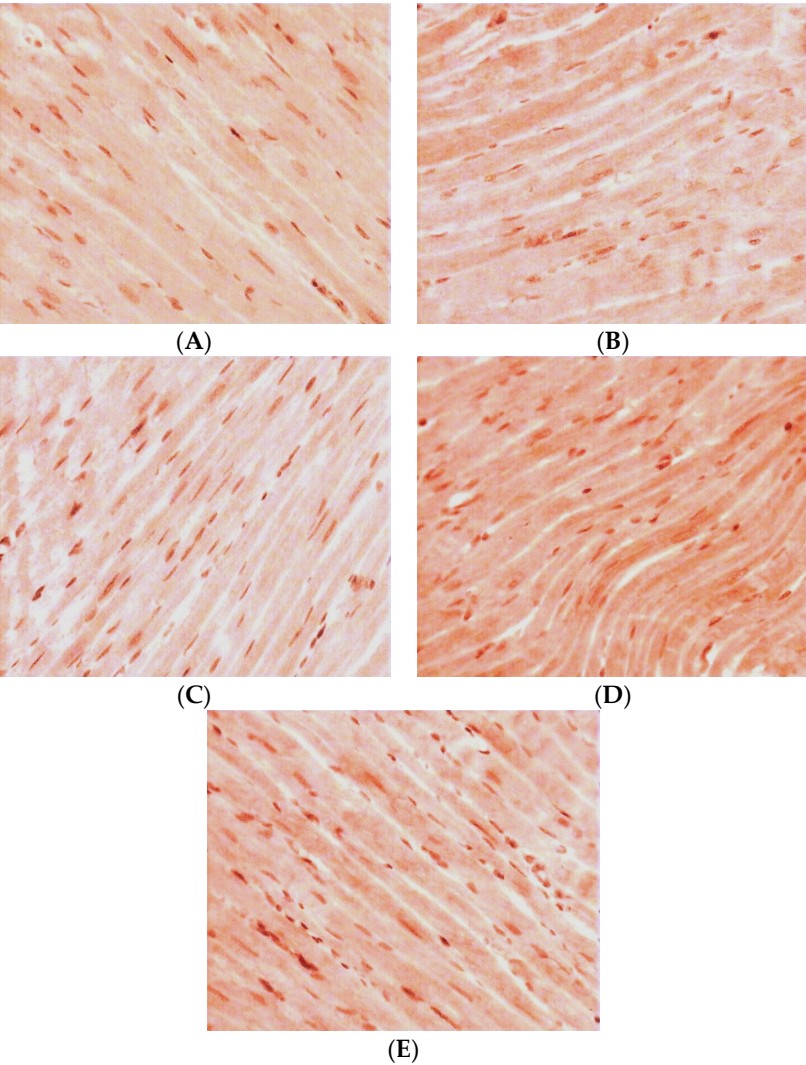

**Figure 4.** Expression of GS in LV CMCs (the middle layer of the myocardium). Immunohistochemical staining, ×400. (**A**) control; (**B**) AH (38-week-old SHR rats); (**C**) AH (57-week-old SHR rats); (**D**) DM (38-week-old WKY rats with insulin-dependent DM); (**E**) AH+DM (38-week-old SHR rats with insulin-dependent DM).

## 4. Discussion

As a result of our experiment, it was found that the content of TXNIP in the cytoplasm of CMCs of the LV myocardium significantly increased compared with the control of isolated DM in the hypertensive of a longer duration (in 57-week-old rats) and in the case of a combination of hypertension with DM. It is interesting that in younger hypertensive animals (aged 38 weeks) the TXNIP level was lower than in the control group. The expression of GS was more than two times higher in the group of isolated DM compared with the controls, and in the case of a combination of hypertension and DM, it was higher than in the control group but lower than in the isolated DM group.

A pronounced increase in TXNIP expression in the DM can be most likely explained by the fact that TXNIP is one of the key factors in glucose metabolism control. For instance, it was shown that TXNIP expression is upregulated by glucose and downregulated by insulin in human muscles and cultured adipocytes [32]. Genetic ablation of the *TXNIP* gene in mice was accompanied by an increase in tissue insulin sensitivity and by enhanced protection against diet-induced resistance to insulin and consequently type 2 DM [32]. In

addition, a study by C. Huang et al., (2016) showed that excessive production of mitochondrial ROS (mtROS) led to the dissociation of TXNIP from its binding protein TXN, which then binds to inflammasome NLRP3 and induces its activation [33,34]. It was also reported that the production of ROS derived from NADPH oxidase (NOX4) contributed to the dissociation of TXNIP-TXN and increased the combination of TXNIP with NLRP3 both in vivo and in vitro. TXNIP may also be involved in inflammation by activating the NLRP3 inflammasome and in this way, it participates in pyroptosis—programmed cell death accompanied by an inflammatory response [35,36].

Enhanced expression of GS in the LV myocardium of the animals with isolated DM is presumably due to the fact that in mammalian cells, the cytosolic and mitochondrial systems of TXN, together with the glutathione system, control the redox mechanisms of the cell [37,38]. In DM, the number of both protective factors represented by glutathione and aggressive factors, such as TXNIP, increases. This is consistent with another study which demonstrated a compensatory increase in the activity of the components of the glutathione system in male persons suffering from type 1 DM, regardless of the level of albuminuria, in the presence of a close relationship between the glycation index and markers of kidney damage in diabetic nephropathy [39]. Therefore, in the case of DM, the production of proteins in the glutathione system may be enhanced.

The increase in TXNIP expression in hypertension of a longer duration is most likely due to the role of oxidative stress in mediating myocardial cell death in response to ATP deficiency resulting from a hemodynamic overload of the LV. Several studies have shown that increased expression of TXNIP during I/R contributed to autophagy-associated CMC apoptosis by upregulating autophagosome formation and inhibiting autophagosome clearance [21,40,41]. Enhanced production of TXNIP is induced by various types of cellular stress, including ischemia [42,43]. In some works, TXNIP is characterized as a mediator of mitochondrial death, which is obviously followed by an ATP deficit and consequent cell death [44–46].

Regarding the increase in GS expression in the LV myocardium of hypertensive rats aged 57 weeks, this may indicate that under long-term hemodynamic overload, the activation of protective mechanisms persists which is manifested as a stimulation of glutathione production. The beneficial role of glutathione in the prophylaxis of hypertension may be explained by the observation that higher levels of glutathione prevent the platelet-derived growth factor (PDGF) mediated production of ROS by NADH/NADPH [47,48]. This indicates that glutathione and NADPH oxidase activity support the redox state and control the production of ROS in mitochondria in hypertension [48]. Therefore, the endogenous antioxidant protection of the myocardium remains even under long-term alteration caused by hemodynamic overload, as evidenced by the enhancement of GS production.

In the case of myocardial damage caused by a combination of hypertension and DM, the level of TXNIP and GS was lower than in isolated hypertension or DM, however, it remained higher in comparison with the control group. These findings can be associated with severe oxidative stress resulting from ATP deficiency due to the summation of metabolic injury and hemodynamic overload. In this case, the decrease in the content of the corresponding markers could have taken place due to a reduction in the number of healthy CMCs, in which TXNIP and glutathione could have been synthesized properly. To some extent, this idea is supported by B. Luo et al., (2014) who demonstrated the role of inflammasome NLRP3 and pyroptosis in the development of diabetic cardiomyopathy. In diabetic rats, there were severe metabolic disturbances, myocardial inflammation, cell death, fibrosis and excessive activation of NLRP3, while NF-κB and TXNIP mediated activation of ROS-induced caspase-1 and IL-1β, which are effectors of inflammasome NLRP3 [49]. Another study showed the role of TXNIP as a therapeutic target for the treatment of myocardial hypertrophy by attenuating oxidative stress caused by TXNIP activation through its binding and TXN release [50]. Hence, inhibition of TXNIP expression, as well as an enhancement of intracellular glutathione production, might contribute to maintaining healthy myocardial mass by reducing oxidative stress in CMCs which is known

to provoke various types of programmed cell death including apoptosis, autophagy, mitophagy and pyroptosis and others.

## 5. Conclusions

According to the obtained data, we can conclude that metabolic disorders of the LV myocardium caused by insulin-dependent DM as well as in a combination of DM with a hemodynamic hypertensive overload of the LV are accompanied by activation of the mechanisms of both oxidative stress and antioxidant protection, which is evidenced by an enhancement of TXNIP and GS production. The results obtained make it possible to use TXNIP as a marker of oxidative myocardial damage and GS as a marker for cardioprotection during insulin-dependent DM and arterial hypertension, as well as use the presented data as a basis for a more detailed study about the mechanisms associated with these markers for both myocardial alteration and cardioprotection. Increases in the expression of GS is a good prognostic sign that even in conditions of severe energy deficiency, the remaining mass of healthy CMCs continues to protect cells from death, and the study of other ways to increase the expression of endogenous glutathione can improve the condition of the myocardium and slow the progression of heart failure. However, there are other markers that will complement the picture of the state of redox-sensitive systems on the pathology models of insulin-dependent diabetes mellitus, as well as myocardial hypertrophy caused by arterial hypertension. Additional data will allow a comprehensive approach to the issue of improving the state of the myocardium in diabetes and hypertension and protecting it from complications.

**Author Contributions:** Conceptualization, A.S. and M.B.; methodology, A.S. and M.B.; software, V.G.; formal analysis, M.A. and V.G.; investigation, A.S and M.B.; resources, M.B.; data curation, M.A.; writing—original draft preparation, A.S.; writing—review and editing, M.B.; visualization, A.S.; supervision, M.B.; project administration, M.B.; funding acquisition, M.B. All authors have read and agreed to the published version of the manuscript.

**Funding:** This publication has been supported by project № 032532-0-000, RUDN University.

**Institutional Review Board Statement:** This study was conducted in accordance with the European Convention for the Protection of Vertebrate Animals used for Experimental and Other Scientific Purposes (Strasbourg, 18.III.1986); the animal study protocol was approved by the Ethical Committee of the RUDN Institute of Medicine.

**Informed Consent Statement:** Not applicable.

**Data Availability Statement:** Data are contained within the article.

**Conflicts of Interest:** The authors declare no conflicts of interest.

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
