# Peer review of "Myocardial Glutathione Synthase and TRXIP Expression Are Significantly Elevated in Hypertension and Diabetes: Influence of Stress on Antioxidant Pathways"

_pathophysiology, doi:10.3390/pathophysiology30020021_

Round 1
Reviewer 1 Report
This paper is focused on antioxidant protection as one of the key reactions of cardiomyocytes in response to myocardial damage. In particular the study evaluates the myocardial Glutathione Synthase and TRXIP expression that are significantly elevated in hypertension and diabetes. The topic is very interesting and original and it is very well written. Only minor revision are needed:
· please change in the title “Glutathioine” in “Glutathione”.
· authors provide information exclusively regarding nutritional regiment that DM rats received, as mentioned in line 100: “a diet adapted for diabetic animals”. However, providing more details about the differences between the above-, mentioned diet and the other feeding regimens would be interesting.
· although it is not the primary aim of the manuscript, this could be also important to possibly obtain more data about correlation between nutrition and metabolic disorder, ROS production and antioxidant responses, cellular and molecular mechanisms.
Minor editing of English language required
Author Response
Dear Reviewer! Thank you so much for the careful analysis of our paper and useful remarks which, we hope, enabled us to improve it. We tried to give the most accurate answers for each of the points of your comments.
Point 1: Please change in the title “Glutathioine” in “Glutathione”.
We have corrected the word.
Point 2: Authors provide information exclusively regarding nutritional regiment that DM rats received, as mentioned in line 100: “a diet adapted for diabetic animals”. However, providing more details about the differences between the above-, mentioned diet and the other feeding regimens would be interesting.
We have added additional information about diet for diabetic animals and the other feeding regimens for rats without DM:
The rats with DM received a diet adapted for diabetic animals (Dry complete diet food PRO PLAN ® Veterinary Diets DM St/Ox Diabetes Management for adult cats to regulate glucose intake (in diabetes mellitus) with low levels of sugars (mono- and disaccharides). Other animals received regular adapted rat food: standard rodent chow diets obtained from Purina LabDiet(®).
Point 3: Although it is not the primary aim of the manuscript, this could be also important to possibly obtain more data about correlation between nutrition and metabolic disorder, ROS production and antioxidant responses, cellular and molecular mechanisms.
Dear Reviewer, unfortunately, the purpose of our study was not to study the correlation between nutrition and metabolic disorders of the body, the production of ROS and the antioxidant response of the body depending on nutrition, as well as the change in the studied parameters depending on the change in the type of nutrition for one group. But in our recent studies (the data has not yet been processed), we led diabetic animals on ordinary feed for laboratory rats and noticed that the life expectancy of such a group was less than in the presented study, that is, they did not live up to 30 days, the average life expectancy of such animals was 18-20 days. Therefore, in the coming articles we will definitely make this correlation.
Reviewer 2 Report
The manuscript from Sklifasovskaya et al. gives an insight into the role of TXNIP and GS, two crucial components of redox-thiol systems, in oxidative myocardial damage and antioxidant protective response induced by the onset of DM, hypertension and combined hypertension and DM. The reviewer believe that the manuscript could be of interest for the readership of the journal, the text is fluent, contains all the key-elements according to the journal requirements. Therefore, the manuscript can be recommended for publication provided the minor revisions outlined below have been adequately addressed by its authors.
1) Lanes 100-101: the authors should specify which diet is appropriate for diabetic animals.
2) Lanes 103-104: the authors should indicate more details regarding the approval of Ethical Committee
3) Paragraph 2.2
- On what basis was the 30-day limit set for the condition of diabetes mellitus before sacrificing animals?
-The authors should clarify why there is a group 3 comprising 57-week-old hypertensive animals and no corresponding control groups of: normotensive, diabetic and normotensive and diabetic and hypertensive. How were these groups chosen? What is the meaning of group 3?
4) The statistical analysis has to be modified by using a non-parametric test for comparison between several groups followed by a post-hoc test to make specific comparisons between groups. The use of the Mann Witheny is incorrect. Consequently, the figures, the text of the results and the discussion must be updated.
5) Did the authors detect morphological changes as an effect of the pathologies studied (ref. paragraph 2.4)? Because only the qualitative aspects of the observed staining are described in the results section, but no morphological analysis is present.
Are there figures for this? In the materials and methods section, it is stated that 'Histological sections (5 μ) were mounted on ordinary glasses for the morphological analysis'.
The two paraphrases should be implemented by adding this information.
6) The figures 2 and 4 (A,B,C,D,E) must be made more comprehensible even to readers who are not experts in immunohistochemical analysis, e.g. by inserting marks that identify the epicardium, endocardium, etc., as well as any other distinctive morphological features induced by pathology.
7) Sentence lanes 309-311: “In this case, the decrease in the content of the corresponding markers could have place due to a reduction of the number of healthy CMCs, in which TXNIP and glutathione could have been synthesized properly.
Do the data reported by the authors support this explanation? How are the number of healthy CMCs determined?
8) The discussion should be improved in the light of the previous comments (3,4,5) and in relation to the pathologies examined, their duration, etcc and if and when these two proteins can be used as reliable markers of oxidative damage.
Author Response
Dear Reviewer! Thank you so much for the careful analysis of our paper and useful remarks which, we hope, enabled us to improve it. We tried to give the most accurate answers for each of the points of your comments.
Point 1: Lanes 100-101: the authors should specify which diet is appropriate for diabetic animals.
We have added additional information about diet for diabetic animals and the other feeding regimens for rats without DM:
The rats with DM received a diet adapted for diabetic animals (Dry complete diet food PRO PLAN ® Veterinary Diets DM St/Ox Diabetes Management for adult cats to regulate glucose intake (in diabetes mellitus) with low levels of sugars (mono- and disaccharides). Other animals received regular adapted rat food: standard rodent chow diets obtained from Purina LabDiet(®).
Point 2: Lanes 103-104: the authors should indicate more details regarding the approval of Ethical Committee.
We have added additional information about Ethical Committee:
The studies were also approved by the Ethical Committee of the RUDN Institute of Medicine (An ethical code number 26, date: February 18, 2021).
Point 3: Paragraph 2.2
On what basis was the 30-day limit set for the condition of diabetes mellitus before sacrificing animals?
The 30-day limit set for the condition of diabetes mellitus was chosen since it was by this time that the animals were weakening and could die from complications caused by diabetes mellitus. Diabetes in animals was not stabilized by insulin therapy, only a glucose-maintaining diet was used. Thus, the duration of diabetes mellitus for 30 days was optimal for occurrence microvascular complications necessary for the study of heart pathology, but life-threatening complications have not yet appeared.
The authors should clarify why there is a group 3 comprising 57-week-old hypertensive animals and no corresponding control groups of: normotensive, diabetic and normotensive and diabetic and hypertensive. How were these groups chosen? What is the meaning of group 3?
The group 3 was chosen in order to compare 2 groups of animals (group 2 and group 3) with different duration of hypertension and analyze how different duration of hypertension affects the state of redox-sensitive systems, because morphologically the difference was in the state of the myocardium. That is, this is a comparison group for a lower period of arterial hypertension. We did not make a control group for 57-week-old normotensive animals, because there was no difference in the state of the myocardium between WKY rats aged 38 weeks and WKY rats aged 57 weeks normotensive rats. We did not make a group of animals with diabetes mellitus and 57-week-old hypertensive animals because the purpose of our study was not to compare groups of animals with arterial hypertension of different duration and induced diabetes mellitus. But in the future, this comparison would be interesting, since groups of hypertensive SHR rats aged 38 weeks and hypertensive SHR rats aged 57 weeks showed different results of the state of redox-sensitive systems in response to hemodynamic overload of different duration. Therefore, in the next studies we will definitely add this group for comparison.
Point 4: The statistical analysis has to be modified by using a non-parametric test for comparison between several groups followed by a post-hoc test to make specific comparisons between groups. The use of the Mann Witheny is incorrect. Consequently, the figures, the text of the results and the discussion must be updated.
We use Mann Witheny test, because it is a non-parametric test for comparison between several small groups and allows you to identify differences in the parameter value. In our study, we compared groups in pairs, for example, different periods of hypertension with each other or a group of diabetes mellitus with control, as well as a group of hypertension with diabetes mellitus with diabetes mellitus separately and separately with a group of isolated hypertension. Thus, for these purposes, we chose the nonparametric Mann-Whitney test. ANOVA analysis or post hoc analysis is used to check the difference between two or more averages. This was not the purpose of our study, since we compared groups in pairs, but in the future, we will definitely try to use this test to compare three or more groups.
Point 5: Did the authors detect morphological changes as an effect of the pathologies studied (ref. paragraph 2.4)? Because only the qualitative aspects of the observed staining are described in the results section, but no morphological analysis is present.
Yes, we have done morphological analysis and post the article previously with this data. (Blagonravov, M. L. Heat shock protein HSP60 in left ventricular cardiomyocytes of hypertensive rats with and without insulin-dependent diabetes mellitus / M. L. Blagonravov, A. P. Sklifasovskaya, A. Yu. Korshunova, M. M. Azova, A. O. Kurlaeva // Bulletin of Experimental Biology and Medicine. – 2020. – Vol. 170. – N1. – P. 10-14.). In this article, morphological analysis was described qualitatively.
Are there figures for this? In the materials and methods section, it is stated that 'Histological sections (5 μ) were mounted on ordinary glasses for the morphological analysis'.
We also have quantitative data for morphological analysis.
The two paraphrases should be implemented by adding this information.
Histological sections were examined using the Nikon Eclipse E400 light microscope at 400x magnification and the TauVideo video system with the Tau Morphology program based on the Watec 221s camera. Morphometric analysis was performed in each field of vision (30 visual fields were analyzed in each myocardial preparation): the relative content of myofibrils, nuclei, vessels, destruction sites and intercellular spaces in the myocardium was determined in volume percentages (vol. %) using the Avtandilov grid [Avtandilov G.G., 1990]. The "nuclear-cytoplasmic ratio" was also calculated: the percentage ratio of the area of CMC nuclei to the area of muscle fibers.
According to morphological analysis, in the group of SHR rats aged 38 weeks, compared with the control group, a significant increase in the volume area of myofibrils and a decrease in the number of nuclei were revealed. Also, hypertrophy of muscle fibers is indicated by a decrease in NCR compared to the control. In the group of SHR rats aged 57 weeks, NCR is reduced relative to the control, which indicates hypertrophy of muscle fibers, however, it increases slightly relative to the group of 38 weeks, which is consistent with the results of previous studies: with long-term hypertension (more than a year), NCR does not differ significantly from the control group (Blagonravov M.L. et al., 2016). In the DM group, the volume fraction of CMCs decreases due to their hypotrophy, while the NCR (nuclear-cytoplasmic ratio) increases. In the AG group of 38 weeks, CMC death occurs in combination with DM, as evidenced by a sharp decrease in the number of nuclei, a return of the percentage of CMC to the control level and a decrease in NCR by 2 times relative to the control.
Point 6: The figures 2 and 4 (A,B,C,D,E) must be made more comprehensible even to readers who are not experts in immunohistochemical analysis, e.g. by inserting marks that identify the epicardium, endocardium, etc., as well as any other distinctive morphological features induced by pathology.
We completely agree. Our pictures demonstrate the middle layer of the myocardium, unfortunately we could not take a photo of all the layers because then the camera resolution would not allow us to see immunohistochemical staining, and the purpose of the picture was precisely to show the characteristic distribution of staining for each group.
Point 7: Sentence lanes 309-311: “In this case, the decrease in the content of the corresponding markers could have place due to a reduction of the number of healthy CMCs, in which TXNIP and glutathione could have been synthesized properly.
Do the data reported by the authors support this explanation? How are the number of healthy CMCs determined?
This hypothesis was based on the obtained morphological analysis data based on a decrease in NCR by 2 times in comparison with the control group. (In the AG group of 38 weeks, CMC death occurs in combination with DM, as evidenced by a sharp decrease in the number of nuclei, a return of the percentage of CMC to the control level and a decrease in NCR by 2 times relative to the control.)
Point 8: The discussion should be improved in the light of the previous comments (3,4,5) and in relation to the pathologies examined, their duration, etcc and if and when these two proteins can be used as reliable markers of oxidative damage.
We have added this information in conclusion.
Reviewer 3 Report
1. In L65 authors have cited the "C. Gao et al. 2020" change into " Gao et al. 2020"
2. Authors have to provide the reference studies of the following sections; 2.3 and 2.4.
3. In L126 place change this "(5 μ)" into " (5μ)".
4. Conclusion size is too short please increase its size by the addition of recommendations.
5. Authors have to provide the strengths and limitations of the study.
I suggest a correction by professional English correction service
Author Response
Dear Reviewer! Thank you so much for the careful analysis of our paper and useful remarks which, we hope, enabled us to improve it. We tried to give the most accurate answers for each of the points of your comments.
Point 1: In L65 authors have cited the "C. Gao et al. 2020" change into " Gao et al. 2020"
We corrected this mistake.
Point 2: Authors have to provide the reference studies of the following sections; 2.3 and 2.4.
We have added the reference studies.
Point 3: In L126 place change this "(5 μ)" into " (5μ)".
We corrected this mistake.
Point 4: Conclusion size is too short please increase its size by the addition of recommendations.
We have added the recommendations in conclusion.
Point 5: Authors have to provide the strengths and limitations of the study.
We have specified the strengths and limitations of the study and added it in conclusion.